

# Effects of fertilizer application on the growth of *Stranvaesia davidiana* D. seedlings

Xue-Man Wang[1,*], Yu-Tong Zhu[2,*], Juan Wang[2], Shi-Hui Wang[2], Wen-Qian Bai[1], Zhi-Fei Wang[2], Wan-Qing Zeng[2] and Pei-Hao Peng[1]

[1] Chengdu University of Technology, College of Earth Sciences, Chengdu, Sichuan, China
[2] Chengdu University of Technology, College of Geography and Planning, Chengdu, Sichuan, China
[*] These authors contributed equally to this work.

## ABSTRACT

Wild plants represent a potential source of urban landscape trees. *Stranvaesia davidiana* Dcne. is a member of the *Stranvaesia* Lindl. Genus, which belongs to family Rosaceae Juss. It has great ornamental value. It can contribute to urban color foliage and fruit species. However, the most effective fertilizer application strategy required for its cultivation is unknown. Therefore, we conducted an orthogonal experiment to investigate the fertilizer type and level (pure nitrogen) using ten experimental groups, including an untreated control group. Pot experiments were used to determine the growth indices of seedlings, including plant height, basal diameter, and chlorophyll content post-fertilizer treatment. This study explored the most appropriate fertiler application model for the growth of *S. davidiana* seedlings. The results revealed that enhanced seedling growth depended on the type and amount of fertilizer used, and their interaction. Fertilizer application increased the plant height by 2.67 cm to 12.26 cm, basal diameter by 0.39 cm to 0.75 cm, and chlorophyll content by 5.66 to 19.86. Among the different types of fertilizer, organic fertilizer increased the plant height by 0.42 cm to 9.59 cm and basal diameter by 0.01 cm to 0.05 cm, compared with the control group. Organic fertilizer had the maximum effect on seedling growth, especially at medium levels. The total growth of basal diameter and chlorophyll content was 1.58 ± 0.04 cm and 39.53 ± 2.37, respectively. Basal diameter is the most critical index in seedling reproduction . The study results suggest that the application of 4.06 g of organic fertilizer per plant was the most effective, and served as a basis for further field trials.

# INTRODUCTION

Wild ornamental plants survive in the wild without domestication. They exhibit excellent ornamental characteristics that can be used to supplement the resource pool for urban landscaping (*Li et al., 2019b*). Advances in urban greening have enriched plant diversity and facilitated seasonal changes in landscape construction (*Zou et al., 2007*). Nonetheless, few types of woody plant species have been utilized including highly colorful and ornamental wild species (*Zhu & Wang, 2011*). In China, a wide range of wild species display the potential

Corresponding author
Juan Wang,
id.wangjuan@foxmail.com

for development of green landscape, but are inadequately utilized currently (*Deng, 1996*; *Zhang, Sun & Shi, 2002*). The introduction and reproduction of local species to support landscape development is cost-effective and has a positive impact on efficient landscape and local biodiversity conservation (*Cai, Xu & Ding, 2020*; *Hu et al., 2010*; *Jiang et al., 2009*). However, the successful introduction and domestication of wild ornamental plants is based on substantial studies including wild surveys, chamber, and field experiments.

*Stranvaesia davidiana* Dcne. is a shrub or small tree belonging to family Rosaceae Juss., with both ornamental and nutritional value (*Liu, Chen & Chen, 2016*; *Wang, 2007*). It possesses a plump crown, bright green leaves that turn red in autumn. The plant is persistent with white compound corymbs and orange-red sub-globose fruits. The plant flowers between May and June, with fruiting period between September and October, with a variety of ornamental flowers, foliage, and fruits (*Yu, 1974*). Its introduction can compensate for scarce woody species in urban gardens due to its colorful leaves and fruits. The plant is widely distributed on slopes, mountain tops, roadsides, thickets, river valleys, and damp gullies at a height of 1000 m to 3000 m. It has diverse habitats and strong adaptability. Thus, it is a typical local species for plant breeding and commercial application. Further, *S. davidiana* has a strong potential for development in bonsai. The fruits have high nutritional value with key functional ingredients (*Liu, Chen & Chen, 2016*; *Wang, 2007*). However, fewer studies have investigated its potential. Studies involved primarily rootless test tube seedlings and germination of the variant *S. davidiana* var. *undulata* (Dcne.) Rehd.&Wils., and its sexual propagation (*Jiang et al., 2009*; *Li et al., 2005*). The absence of studies investigating its growth mechanisms and adaptation has constrained its large-scale production and commercial application. Preliminary studies involved seed germination (*Wang et al., 2020*) under simulated fertilizer application and determination of soil nutrient status under different environmental conditions *via* pot experiments. The growth performance and adaptation of seedlings under experimental conditions provided insight into the nutritional requirements and growth patterns for efficient cultivation.

Fertilizer treatment is an effective strategy that can improve the quality of seedling growth (*Chen et al., 2012*; *Zheng et al., 2016*). Scientific and rational application of fertilizers can be used to regulate soil nutrient levels rapidly, increase plant nutrient demand, promote seedling growth, and accelerate the synthesis and accumulation of plant metabolites. Fertilizers can be used to replenish mineral ions in the soil for the cultivation of terrestrial species (*Martínez-Sánchez, 2006*), thereby increasing the rate of plant photosynthesis, height, basal diameter, and volume (*Büyük et al., 2022*; *Lincoln et al., 2007*; *Mancus, 2007*). Application of large amounts fertilizers irrationally can lead to soil degradation, decline of soil organic carbon content, and destruction of soil structure (*Galloway et al., 2008*; *Koch et al., 2021*; *Mancus, 2007*; *Paungfoo-Lonhienne et al., 2019*). The type and amount of fertilizer applied do not meet the demands of plant growth, which can result in low yield and quality, and ultimately affect the production (*Kentelky & Szekely-Varga, 2021*; *Wilber & Williamson, 2008*). Studies show that a wide range of models are not applicable to all plants, fertilizers, and regions (*Konde et al., 2009*; *Osmond & Riha, 1996*). The optimal fertilizer treatment pattern for *S. davidiana*, and its response to different nutrient levels is not clear. Therefore, determining the optimal fertilizer type and dosage under controlled conditions *via* pot
experiments is of great significance in advancing field trials and large-scale production (*Bai, 2015*).

This study used first-year seedlings of *S. davidiana* to analyze the impact of different fertilizers such as organic, inorganic, and nitrogen fertilizers and their levels based on pure nitrogen level on growth indicators. The relevant breeds were treated with the standard fertilizer (pure N) per acre in potted containers based on soil volume. The low N level was based on a standard N application of one-half amount, while a high N level was based on double the amount. An orthogonal experiment was set up combined with the fertilizer type. The control group was not treated with a fertilizer. Our goal was to explore the most appropriate model of fertilizer treatment of *S. davidiana* seedlings *via* comparative experiments. The results are expected to provide a reference for seedling management, improve their quality, guide cultivation, and facilitate marketing and large-scale production.

## MATERIALS & METHODS

### Experimental field and materials

The experiment was carried out in the internship base of Chengdu University of Technology, Chengdu City, Sichuan Province, with a geographic location of 30°40′43.50″E and 104°08′20.15″N, and an elevation of 512 m. The location has a subtropical monsoon climate, with an average annual temperature of 15.9 °C. The hottest months were July and August, with an average monthly temperature of 25.2 °C. The coldest month was January, with an average monthly temperature of 5.6 °C. The annual precipitation was 900 mm to 1300 mm. The average annual relative humidity was 82%. The experiment was set up in a greenhouse with stable temperature and humidity from April to December 2019.

*S. davidiana* seedlings were sown in April 2018 using seeds obtained from Tangjiahe National Nature Reserve. The plant materials were identified by plant ecology expert Prof. Peng Peihao at Chengdu University of Technology. The selected seedlings carried well-developed root systems, closed growth, and no pests or diseases. Before transplanting, the original soil was watered adequately to facilitate the removal of intact plants. We used plastic seedling pots measuring 15 cm in diameter and 12.5 cm in height, with neutral filter paper at the bottom to prevent soil loss. The mixture of yellow loam and fine river sand (mixing ratio: 1:1, Ph: 6.9) was used as a substrate based on pre-experimental analysis. Seedlings were transplanted on April 30, 2019 (1 plant/pot), with a mulching height 2 cm from the edge. The seedlings were watered moderately after transplanting and treated with fertilizer a week later to allow seedling restoration.

### Experimental design

In the absence of established fertilizer treatment for *S. davidiana,*, the optimum N application during the year was estimated at 45 to 90 kg/hm$^2$ based on related studies and field experiments (*He et al., 2014*). In this study, the standard amount was 90 kg/hm$^2$. Based on conversion by area, the reference annual fertilizer application was 0.203 g/plant. The group treated with Stanley brand inorganic fertilizer (15% N, 15% phosphorus, and 15% potassium) was designated as SF. The plants treated with Stanley brand organic

**Table 1** The types and amounts of fertilizers used in the experiments, the unit used is g.

| Fertilizer types | Low-level (1) | Medium-level (2) | High-level (3) |
|---|---|---|---|
| SF | 0.677 | 1.354 | 2.708 |
| SY | 2.03 | 4.06 | 8.08 |
| CN | 0.2187 | 0.4374 | 0.8748 |
| CK | 0 | | |

Notes.

SF, inorganic fertilizer; SY, organic fertilizer; CN, nitrogenous fertilizer; CK, no fertilization treatment.

1, 2, and 3 represent the different fertilization levels according to the pure N setting.

fertilizer (45% organic matter, 5% N and 5% P and 5% K) were grouped under SY, and those exposed to Yuzhu brand urea (46.4% N) as CN. Three levels, *i.e.,* one-half of the standard amount (low: 0.1015 g), standard amount (medium: 0.203 g), and double the standard amount (high: 0.406 g) were set and represented by numbers 1, 2, and 3, respectively, as shown in Table 1. The control group (CK) was not treated with any fertilizer. A total of ten experimental groups were set, with 20 replicates each.

To avoid burning seedlings due to excessive fertilizer application, we divided the fertilizer amount into three portions, applied on May 5, July 5, and September 5, 2019, respectively. The soil was loosened and the fertilizers were applied slightly away from the roots. They were watered moderately, away from the hot sun. During the experiment, the plants were watered every two days to ensure consistency of moisture and light between different groups (*Chen et al., 2016*).

## Data recording and analysis

To prevent observational errors associated with recording, we used fixed observers and standardized procedures. The plant height was measured along a straight edge. The vertical height from the ground to the terminal bud was measured. The basal diameter was measured one cm above the topsoil with a digital vernier calipers (accuracy of 0.01 mm) along two vertical directions at the rhizome and then averaged. The chlorophyll content was measured with a hand-held chlorophyll tester (SPAD-502, TYS-A), marking one of the 3rd to 5th healthy leaves under the terminal bud. The leaf vein was avoided. Three points were selected on both sides and tips of the leaves, and the average value was determined each time. The observations were carried out from May to December 2019. All seedlings were measured and recorded individually at the end of each month. The experimental data were organized in Excel 2013, followed by analysis of variance (ANOVA) and multiple comparisons using SPSS 21.0. The data were plotted using Origin 2022.

## RESULTS

### Effect of different fertilizer types on *S. davidiana* growth

Organic fertilizer had the most remarkable effect on the growth of seedlings at similar levels of application. Compared with other groups, the SY treatment increased the plant height by 0.42 cm to 9.59 cm and the basal diameter by 0.01 cm to 0.22 cm, excluding SPAD values. Specifically, the SPAD value increased in the SY group by 4.38 to 2.14 in low

and medium levels compared with other groups. The chlorophyll content of SF3 (34.58) and CN3 (33.54) was better than in SY3 (25.33) at high levels.

The inter-monthly variability in plant height and the monthly peak values were inconsistent across different fertilizers types, even though the growth was progressively lower in all experimental groups as the months increased and decreased. SY species were the most effective, with their peaks occurring in May. The growth declined over time, although a small peak was detected in July under medium and high levels of treatment. Compared with organic fertilizers, the peaks in SF and CN groups were detected in June and were particularly pronounced under high levels, with a monthly growth of 11.43 ± 0.62 cm following treatment with inorganic fertilizer and 12.75 ± 0.58 cm using nitrogen fertilizer. Under low levels, the CN group showed a lower monthly growth in August and September than the CK group, albeit insignificantly (0.01–0.14 cm). The base diameter variation showed a contrasting trend. Regardless of the type, the peak growth occurred in June and was inconsistent at different levels of treatment. The CN1 group (0.59 ± 0.03 cm) outperformed SY1 (0.54 ± 0.02 cm) and SF1 (0.45 ± 0.02 cm) at low levels, whereas SY2 (0.67 ± 0.03 cm) exceeded SF2 (0.57 ± 0.09 cm) and CN2 (0.56 ± 0.04 cm) under medium level. Under high levels, the performance of SF3 (0.64 ± 0.03 cm) and CN3 (0.64 ± 0.03 cm) was comparable and higher than that of SY3 (0.52 ± 0.02 cm). The SPAD values peaked in all experimental groups in June, far exceeding those of the control group (9.71–19.58). They decreased rapidly in July, and resembled those of the CK group or even slightly lower (−2.83−1.56). Although the SPAD values fluctuated across different months and were even often lower than the CK values, they tended to be almost negative in November (except for SY3, with a value of 0.09). The chlorophyll content in November was higher in the SY group than in SF and CN, and all were significantly higher than in the CK group. Additional details are discussed in Fig. 1 and the Appendix.

### Effect of different fertilizer levels on *S. davidiana* seedling growth

Increased level of similar fertilizer did not always enhance seedling growth. Treatment with increasing levels organic fertilizers decreased the plant height eventually: SY1 (9.89 ± 0.42 cm) >SY2 (9.13 ± 0.51 cm) >SY3 (9.78 ± 0.50 cm). The effect of medium level was also superior to high-level treatment on both basal diameter and SPAD values. A similar phenomenon was observed in the SPAD values of nitrogenous fertilizers. Similar results were obtained in 45% of the experimental groups in our study, while the remaining 55% showed that high-level treatment enhanced the growth of *S. davidiana*.

The analysis of plant height in both SF and SY groups revealed peaks in June and further enhancement with increasing fertilizer amounts. The extreme values (SF3: 11.43 ± 0.62 cm, CN3: 12.75 ± 0.58 cm) were much higher than those of the CK group (6.80 ± 0.47 cm). In the SY group, the peaks were observed in May with slight differences between groups (9.13−9.89 cm) and insignificant difference from the CK (7.56 ± 0.53 cm) in the same month. Based on the observed inter-monthly variations in basal diameter and peaks in June, differences were found in the growth response to fertilizer. Exposure of the SY group to medium level had the strongest effect on basal diameter increase. However, in the CN group, the medium-level treatment had the least effectiveness. The SF group showed
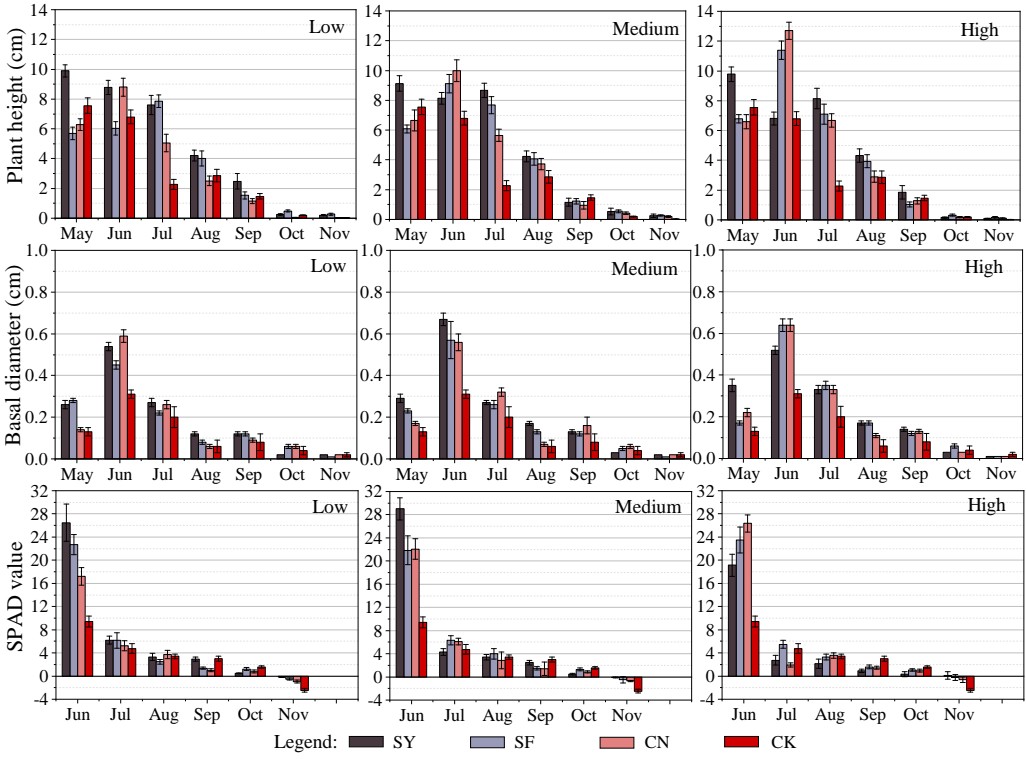

**Figure 1** **Response of seedling growth indicators to different fertilizer types at the same level of fertilization.** Different colors represent different types of fertilizers.

enhanced positive effect under increased levels of application. Peak chlorophyll levels were also observed in June, and the chlorophyll content increased from 7.8 to 17.05 at low levels, 12.40 to 19.58 at medium levels and 9.71 to 16.91 at high levels compared with the CK group. Similarly, the stabilizing effect on chlorophyll irrespective of the fertilizer levels, was significantly superior to the CK In November, chlorophyll content was higher in all experimental groups treated with fertilizer than in the CK group, as shown in Fig. 2 and the Appendix.

## Effect of different fertilizer parameters on *S. davidiana* growth

ANOVA was used to analyze the effect of fertilizer type, amounts, and their interaction on the total growth of *S. davidiana* seedlings. The fertilizer type and amount significantly affected plant height and basal diameter, but to a markedly different extent. Plant height was significantly affected by fertilizer type ($P < 0.01$), while basal diameter was significantly affected by fertilizer concentration ($P < 0.01$). However, the combination of fertilizer type and amount did not have a significant effect on the growth of *S. davidiana* seedlings in terms of plant height and basal diameter, but showed a strongly significant effect ($P < 0.01$) on SPAD values. We also found that the SAPD value was the only parameter that did not exhibit a significant response ($P = 0.29$) to the type of fertilizer in ANOVA (type or amount

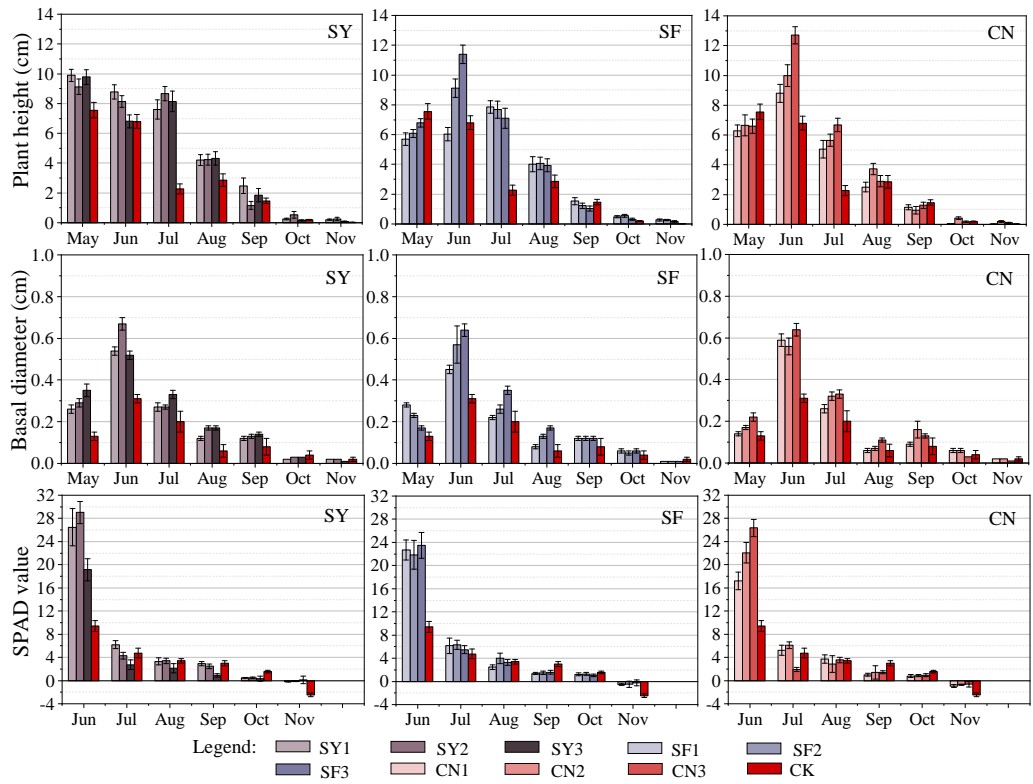

**Figure 2  Response of seedling growth indicators to different fertilizer levels at the same types of fertilization.** Different shades of color represent different levels of fertilization.

**Table 2  Results of analysis of variance (ANOVA) of different fertilization measures on the total growth of *S. davidiana* seedlings.**

| | Fertilization types | | Fertilization amounts | | Fertilization types * amounts | |
|---|---|---|---|---|---|---|
| | F | *P* | F | *P* | F | *P* |
| Plant height | 8.56 | <0.01** | 3.06 | 0.05* | 2.26 | 0.06 |
| Base diameter | 3.64 | 0.03* | 11.81 | <0.01** | 0.58 | 0.68 |
| SPAD values | 1.24 | 0.29 | 3.91 | 0.02* | 3.74 | <0.01** |

**Notes.**
*represents a significant difference at the 0.05 level and ** represents a highly significant difference at the 0.01 level.

of fertilizer). Nevertheless, all seedling growth indicators showed a significant response to fertilizer amount (Table 2).

Despite fluctuations in the monthly growth of these indicators in different months in our study, and even lower levels than in the CK group, in terms of total growth (see Fig. 3), all groups treated with fertilizer exhibited higher growth than the CK group. The total growth of plant height in the experimental groups was significantly higher than in the CK group, except for CN1, which did not appear significant. Further, the total growth in basal diameter and SPAD values of all groups treated with fertilizer were significantly higher than in the CK group. Fertilizer treatment led to an increase in the height of S. davidiana

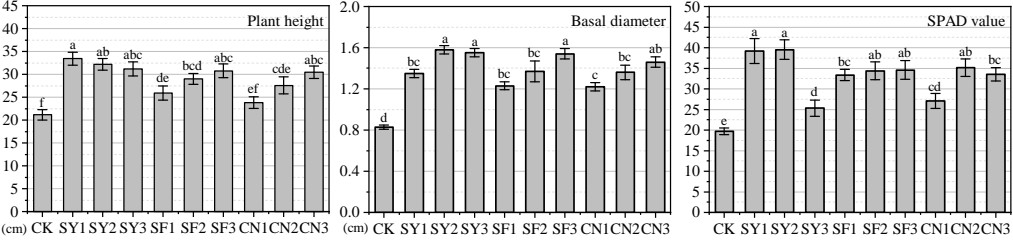

**Figure 3 Changes in total growth of seedling growth indicators in response to different fertilization measures.** Letter labels represent significant differences at the 0.05 level.

seedlings by 2.67 to 12.26 cm, basal diameter by 0.39 to 0.75 cm, and SPAD values by 5.66 to 19.86. The combination of fertilizer type and levels led to a maximum increase in plant height of SY1 (33.42 ± 1.39 cm), but the least in SF1 (25.89 ± 1.57 cm). The SY2 group showed the most increase in basal diameter (1.58 ± 0.04 cm), while the CN1 group showed the least (1.22 ± 0.04 cm). Under similar SPAD values, the SY2 group exhibited the utmost enhancement (39.53 ± 2.37), and SY3 the least (25.33 ± 1.97) seedling growth.

## DISCUSSION

The fertilizer requirement varies with species, mainly depending on the type and amount of fertilizer applied (*Lu et al., 2022*; *Luo et al., 2023*; *Wang et al., 2011*; *Yao et al., 2017*; *Zhao et al., 2020*). Nitrogen is the most essential element for plant growth, contributing 40% to 50% of the required nutrition (*Bown et al., 2010*; *Feng et al., 2021*; *Yang & Wu, 2019*). Its effectiveness and internal concentration affect the biomass partitioning between roots and shoots (*Bown et al., 2010*).The soil N content and duration of action also alter plant morphology and photosynthesis (*Zhao et al., 2008*). Further, N promotes cytokinin production that further affects cell wall elasticity, the number and growth of plant meristem, increases the ground diameter, and improves plant height (*Bloom, Frensch & Taylor, 2006*; *Du et al., 2019*). Chlorophyll, required for plant photosynthesis, is composed of numerous N elements. It promotes the formation of chloroplasts during leaf growth and active photosynthesis, enhancing the photosynthetic efficiency of plants (*Li et al., 2012*). These factors explain the significant growth performance of all the experimental groups than the untreated CK group in our study.

The amount and type of fertilizer used altered the growth response of *S. davidiana* under similar conditions. The highest total increase in plant height and basal diameter of seedlings exposed to organic fertilizer (SY group) is attributed to the abundant organic matter in addition to inorganic elements, which not only provides nutrients for plants but also refines soil enzyme activity and water storage capacity, improves microbial ecology as well as soil physicochemical properties (*Cai, Xu & Ding, 2020*; *Li et al., 2008*; *Li et al., 2019a*; *Oliveira et al., 2022*; *Yang, 1996*), ultimately promoting seedling growth. These results have been validated in previous studies (*Chen et al., 2021*; *He et al., 2014*; *Wang et al., 2023*). Conversely, treatment with inorganic and nitrogen fertilizers (SF and CN groups, respectively) leads to N leaching following watering, due to the lack of organic matter,

resulting in destruction of soil physical structure and sloughing (*Luo et al., 2017*), which affects seedling growth. Uncontrolled increase in the proportion of N does not always enhance seedling growth. However, the rate of chlorophyll decline was diminished with fertilizer treatment compared with the CK group, which effectively delayed leaf senescence and promoted photosynthesis in the plant (*Chen et al., 2016*; *Yang & Wu, 2019*; *Yao et al., 2017*). However, adequate or even excessive growth under P and K deficiency may suggest a relative imbalance of N/P values within the leaves, which affects chlorophyll synthesis (*Hernández Valera, López López & Flores Nieves, 2018*). Thus, the results suggest large variation in SPAD values within the CN compared with relatively stable levels in other groups.

The fertilizer level plays a critical role in altering the plant yield and quality. Insufficient fertilizer application suppresses plant yield and quality. However, over-application results in either environmental pollution or inhibition of nutrient uptake by plant roots as excessive nutrient supply exceeds the saturation state of the soil (*Zheng et al., 2016*). This also explains the significant response of all growth indicators to fertilizer dosage in our experiments. At high levels, the variation in total growth of plant height and basal diameter was relatively minor and close to the maximum regardless of fertilizer type. These findings were consistent with several studies, such as those involving *Yulania sprengeri* (Pampanini) D. L. Fu, which showed a significant increase in the overall biomass following fertilizer treatment (*Deng et al., 2019*). However, the varying levels of fertilizer treatment had diverse effects on seedling growth enhancement. Organic fertilizers increased plant height significantly at low levels but decreased with increasing levels of application. Medium levels promoted growth in basal diameter. Inorganic and nitrogen fertilizers promoted plant height and basal diameter least effectively at low levels and were more effective with increasing levels. It may be related to the particle size and decomposition of different fertilizers, resulting in differential release of available nutrients, as reported previously in many studies investigating the effect of fertilizer decomposition rate on seedling growth (*Fan et al., 2009*; *Tang et al., 2007*). The SPAD values associated with organic and nitrogen fertilizers showed contrasting trends. Increase in the application of nitrogen and organic fertilizers tended to increase and then decrease the SPAD values, indicating that excess nitrogen reduced the chlorophyll content, which was detrimental to the photosynthesis of the seedlings. It has been demonstrated that excess N shortened the life and increase the susceptibility of seedling leaves to disease (*Bojović & Stojanović, 2005*). Another possible explanation is that increase in leaf N content beyond a certain threshold leads to increased nitrogen assimilation, competing with photosynthetic carbon assimilation during photosynthetic light reactions. Thus, the assimilative power is decreased. The enhanced nitrogen assimilation requires large carbon scaffolding, whereas respiration cannot provide adequate N scaffolding (*Araya, Noguchi & Terashima, 2010*; *Daughtry et al., 2000*). Thus, the assimilation rate is low.

Further, the fertilizer application is associated with the plant life cycle (*Du et al., 2019*; *Yao et al., 2017*). In our study, all experimental groups showed faster growth from May to July, especially plant height and basal diameter. The maximum monthly growth in basal diameter of all seedlings occurred in June, with a slow growth over time. The initial application of fertilizer ensured adequate nutrient supply, thereby accelerating the growth

of *S. davidiana* seedlings in the early stages. As the amount of fertilizer was increased, the positive correlation between plant biomass and nutrient supply under nutrient-poor conditions changed to an inhibitory role under excessive levels (*Zheng et al., 2016*), thereby leading to a decrease in the promotional effect of the fertilizers. However, the growth characteristics of the seedlings themselves may be possible factors. Following rapid growth, the seedlings naturally slow down and gradually enter a dormant state to adapt to the upcoming cold environment. This adaptation has been demonstrated in numerous studies of cyclic plant growth. Studies suggest that treatment with fertilizers rationally during the plant growing season maximizes the fertilizer efficiency, promotes plant growth, and improves yield (*Lu et al., 2022*; *Luo et al., 2023*).

# CONCLUSIONS

This study demonstrates the interaction between different fertilizer types and fertilizer levels. Based on the response of seedling height, basal diameter, and chlorophyll content, the study reveals the fertilizer requirements of *S. davidiana* seedlings. The results indicate that both fertilizer type and level have significant effects on seedling growth. The results also suggest that the application of organic fertilizer (4.06 g/plant) promoted optimal growth. Our study provides insights into the fertilizer patterns of *S. davidiana*. The findings advance our knowledge of artificial breeding of *S. davidiana*, an excellent wild ornamental plant species. It serves as a reference for breeding similar species. This will help improve the yield and quality of *S. davidiana*, provide a foundation for subsequent field trials and large-scale production, and ultimately contribute towards its utilization in urban landscapes. We also expect that our findings based on a preliminary study of this woody ornamental species belonging to family Rosaceae will facilitate other related studies in the future.

# ACKNOWLEDGEMENTS

We sincerely thank Ms. Tan Liping, Ms. Xu Qian, Mr. Pang Xin, and Mr. Bai Hai from Chengdu University of Technology for helping us with the experimental operation. We also thank Mr. Liu Xian'an from Sichuan Tourism University for his help in collecting seeds.

## Funding
The Forestry and grassland ecological protection and restoration funds: 80000-23Z05 supported the APC of this article. The funders had no role in study design, data collection and analysis, decision to publish, or preparation of the manuscript.

## Grant Disclosures
The following grant information was disclosed by the authors:
The Forestry and grassland ecological protection and restoration: 80000-23Z05.

## Competing Interests

The authors declare there are no competing interests.

## Author Contributions

- Xue-Man Wang conceived and designed the experiments, performed the experiments, analyzed the data, prepared figures and/or tables, authored or reviewed drafts of the article, and approved the final draft.
- Yu-Tong Zhu conceived and designed the experiments, performed the experiments, analyzed the data, prepared figures and/or tables, authored or reviewed drafts of the article, and approved the final draft.
- Juan Wang performed the experiments, authored or reviewed drafts of the article, and approved the final draft.
- Shi-Hui Wang performed the experiments, analyzed the data, authored or reviewed drafts of the article, and approved the final draft.
- Wen-Qian Bai performed the experiments, prepared figures and/or tables, authored or reviewed drafts of the article, and approved the final draft.
- Zhi-Fei Wang performed the experiments, analyzed the data, authored or reviewed drafts of the article, and approved the final draft.
- Wan-Qing Zeng conceived and designed the experiments, performed the experiments, analyzed the data, prepared figures and/or tables, authored or reviewed drafts of the article, and approved the final draft.
- Pei-Hao Peng conceived and designed the experiments, authored or reviewed drafts of the article, and approved the final draft.

## Data Availability

The raw data are available in the Supplemental Files.

## Supplemental Information

Supplemental information for this article can be found online at http://dx.doi.org/10.7717/peerj.16721#supplemental-information.

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
