# Peer review of "Effects of fertilizer application on the growth of Stranvaesia davidiana D. seedlings"

_PeerJ, doi:10.7717/peerj.16721_

## Round 0.1 · original submission · Major Revisions

Revised the manuscript as per the comments and resubmit for consideration.

**Language Note:** The review process has identified that the English language must be improved. PeerJ can provide language editing services - please contact us at copyediting@peerj.com for pricing (be sure to provide your manuscript number and title). Alternatively, you should make your own arrangements to improve the language quality and provide details in your response letter. – PeerJ Staff

·

Basic reporting

Dear Editor,
After peer review of the MS entitled “Effects of fertilization on the growth of Stranvaesia davidiana seedlings”, I suggest the MS needs major revision. The MS may be revised as per the comments below:
1. In the abstract add a concluding remark for the study.
2. As mentioned in line no.51 ‘the S. group’ is not a proper scientific description. Redraft accordingly.
3. The importance of the plant Stranvaesia davidiana needs to be highlighted in the introduction for which the study has been conducted.
4. The English of the MS needs to be improved by a fluent English speaker.
5. What was the pH of the soil used for the experiment.
6. Mention the frequency of irrigation used in the study.
7. Some of the images provided are not clear. Provide clear images for the same.
8. In table 2, footnote should be provided for the symbol used.
9. The term plant height growth is not appropriate, instead use the term plant height. Rectify throughout the MS.
10. In the result section, the sentence ‘The maximum increase in SPAD……..SF2 (21.83) > CK (9.43)’ the sentence needs to be redrafted so as the meaning for increasing or decreasing effect is clearly revealed. Redraft accordingly.
11. In line no. 198, the sentence should not begin with the figure no. Redraft the sentence accordingly.
12. The result from table no. 2 needs to be elaborated in the result section.
13. The discussion has been elaborated in length. It needs to be concise.
14. The conclusion should include the future prospect of the study as well.

Experimental design

The trials needs to be done in atleast ten replicates.

Validity of the findings

The ANOVA results needs to be elaborated. Moreover, the data needs to be provided with standard deviation.

Reviewer 2 ·

Basic reporting

Dear Editor,
Thank you so much for providing me with the opportunity to review the manuscript “peerj-reviewing-87321” entitled “effects of fertilization on the growth of Stranvaesia davidiana seedlings.” The manuscript examined the effect of organic, inorganic and N fertilizer on seedling of S. davidiana. The whole study is a very well designed one, but it has lots of short comings. Only after addressing the comments the manuscript can be considered for further proceedings. The decision over the manuscript is “Major Revisions”. All the required corrections are highlighted inside the manuscript with attached comment boxes. Authors are asked to go through all of them and correct them in the revised version of the MS.
Comments:
1. Abstract: Wild plantss: It should be "wild plant".
2. Abstract: Page 6, Line number 17: Write down the identifier name, with the scientific name as well. After that author can write all the scientific names in the MS in standard abbreviated styles.
3. Abstract: Page number 6, Line number 28: wasapplied: was applied. Spacing issues are there. Resolve the issue throughout the whole MS.
4. Abstract: seeding growth should be written as "seedling growth".
5. Abstract, Line number 39: Author has written two "The the", why??
6. Abstract: Abstract is okay but it seems to be little more descriptive in nature rathar than being supported by numerical values. Restructure the whole abstract section accordingly.
7. Keywords: Arrange all the keywords in alphabetical order.
8. Introduction: Zou et al., 2007: Author need to re-check all the cited references and must need to present them in accordance to the Journal format only.
9. Introduction: Page number 6, Line number 51: "Meanwhile, the S. group provides"- What does the author meant by "S. group"?? Need explanation in this aspect.
10. Introduction, Page 7, Line number 63: Write all the scientific names in italics form throughout the whole MS.
11. Introduction: Page 7, Line number 77-78: Gap of the research is not properly established in the introduction section of the MS.
12. Introduction: Introduction part needs to be re-constructed, with a proper flow and link between two separate paragraphs. Cite some recent references with proper developments.
13. Experimental field materials: Page 8, Line number 1001-101: By whom exactly the identification was carried out?? Write the name of the identifier with proper affiliation and designation.
14. Page 8, Line number 128-129: burning: sun burning.
15. Page 8, Line number 131: experiment(Chen et al., 2016). Space required.
16. Data analysis: Page 9, Line number 133-142: All these collections of data were carried out in replica or just in an individual plant?? Is there any standard methodology followed or the authors did it by themselves?? Need clarifications in this aspect.
17. Results: Author need to establish the results section with more of a scientific manner. All the results established throughout the manuscript is good but it seems to be little un-organized. Organize all the results in more of a scientific manner.
18. Discussion: Page 13, line number 292-292: Write all the scientific names in italics only.
19. References: All the references must need to be strictly in accordance to the Journal format, only.
20. Figures: All the figures’ resolutions must need to be increased in the revised version of the manuscript.
21. Tables: All the tables need to be self-standing in nature. And additionally, all the used abbreviations of the table must need to be written in full as Table foot note.
22. Additionally, the language of the manuscript must to be rechecked by a language expert before resubmitting. It should be void of any grammatical, or typical errors.

Experimental design

NA

Validity of the findings

NA

Additional comments

NA

Annotated reviews are not available for download in order to protect the identity of reviewers who chose to remain anonymous.

---

## Round 0.2 · Minor Revisions

Revise the manuscript as per the comments of the reviewers and resubmit for consideration.

**Language Note:** The review process has identified that the English language must be improved. PeerJ can provide language editing services - please contact us at copyediting@peerj.com for pricing (be sure to provide your manuscript number and title). Alternatively, you should make your own arrangements to improve the language quality and provide details in your response letter. – PeerJ Staff

·

Basic reporting

After peer review of the revised manuscript I found that the authors have made substantial changes. However, there are some minor revisions required. The MS may be revised in light of the comments below:
1. The novelty of the study needs to be highlighted.
2. The result for the Table 2 needs to be further elaborated.
3. The English of the MS needs to be further enhanced.

Experimental design

It is adequate.

Validity of the findings

A correlation study would help in effective report.

Reviewer 2 ·

Basic reporting

Dear Editor,
Thank you so much for giving me this opportunity to re-review the manuscript “87312-v1” entitled “Effects of fertilization on the growth of Stranvaesia davidiana seedlings.” The manuscript needs some minor corrections. All the required corrections are highlighted inside the manuscript with attached comment boxes. The decision over the manuscript is “Minor Revision”.
Comments:
1. Title: Write the scientific name with the identifier name. For example "Scientific name L.,"
2. Abstract: Avoid the use of words such as we, our, us, I, you etc.
3. Introduction Page No. 56-60: For this whole statement author did not cite any reference why??
4. Conclusions: Line no. 305: Resolve all the spacing issues throughout the manuscript.

Experimental design

NA

Validity of the findings

NA

Additional comments

NA

Annotated reviews are not available for download in order to protect the identity of reviewers who chose to remain anonymous.

---

## Round 0.3 · accepted · Accept

All comments has been resolved properly now the MS is ready for publication